

# Pan-cancer analysis identifies olfactory receptor family 7 subfamily A member 5 as a potential biomarker for glioma

Yanqiu Bao[1,*], Ziqi Tang[2,*], Renli Chen[3], Xuebin Yu[3] and Xuchen Qi[4,5]

[1] Department of Medical Research Center, Shaoxing People's Hospital, Shaoxing Hospital, Zhejiang University School of Medicine, Shaoxing, Zhejiang, China
[2] Institute of Psychology, Chinese Academy of Sciences, Beijing, China
[3] Department of Neurosurgery, Shaoxing People's Hospital, Shaoxing, Zhejiang, China
[4] Department of Neurosurgery, Sir Run Run Shaw Hospital, Zhejiang University School of Medicine, Hangzhou, Zhejiang, China
[5] Department of Neurosurgery, Shaoxing People's Hospital, Shaoxing Hospital, Zhejiang University School of Medicine, Shaoxing, Zhejiang, China
[*] These authors contributed equally to this work.

## ABSTRACT

**Background**. Human olfactory receptors (ORs) account for approximately 60% of all human G protein-coupled receptors. The functions of ORs extend beyond olfactory perception and have garnered significant attention in tumor biology. However, a comprehensive pan-cancer analysis of ORs in human cancers is lacking.

**Methods**. Using data from public databases, such as HPA, TCGA, GEO, GTEx, TIMER2, TISDB, UALCAN, GEPIA2, and GSCA, this study investigated the role of olfactory receptor family 7 subfamily A member 5 (OR7A5) in various cancers. Functional analysis of OR7A5 in LGG and GBM was performed using the CGGA database. Molecular and cellular experiments were performed to validate the expression and biological function of OR7A5 in gliomas.

**Results**. The results revealed heightened OR7A5 expression in certain tumors, correlating with the expression levels of immune checkpoints and immune infiltration. In patients with gliomas, the expression levels of OR7A5 were closely associated with adverse prognosis, 1p/19p co-deletion status, and wild-type IDH status. Finally, *in vitro* experiments confirmed the inhibitory effect of OR7A5 knockdown on the proliferative capacity of glioma cells and on the expression levels of proteins related to lipid metabolism.

**Conclusion**. This study establishes OR7A5 as a novel biomarker, potentially offering a novel therapeutic target for gliomas.

## INTRODUCTION

Glioma, a frequently occurring primary malignant brain tumor, accounts for approximately 81% of all primary malignant brain tumors and has an increasing global prevalence (*Ostrom et al., 2014*). Gliomas are distinguished by their high occurrence rate, frequent relapse, and elevated mortality rate (*Louis et al., 2021*). The blood–brain barrier hinders the penetration

Corresponding author
Xuchen Qi, qixuchen@zju.edu.cn

of numerous chemotherapeutic agents into the brain, greatly impeding their effectiveness. Currently, a multimodal approach encompassing surgical intervention combined with radiotherapy, chemotherapy, and targeted therapies is the principal treatment strategy for patients diagnosed with gliomas (*Yang et al., 2022*; *Zhang et al., 2019*). Furthermore, the pivotal role of molecular diagnostics in augmenting the precision of glioma diagnosis and prognostic treatment responses highlights its importance. Consequently, it is imperative to identify innovative molecular markers to improve the diagnosis and treatment efficacy of gliomas.

Olfactory receptors (ORs) are a category of membrane receptors that possess the ability to detect odor molecules, thereby playing a pivotal role in olfactory perception (*Shum et al., 2015*). Several studies have proposed that ORs are not solely confined to the nasal cavity but are also expressed in various other tissues, including smooth muscle cells, skin cells, and hair follicles (*Edelkamp et al., 2023*; *Huang et al., 2020*; *Kang et al., 2022*). Consequently, they may contribute to the metabolic regulation, hair growth, wound healing, and other physiological processes (*Chéret et al., 2018*; *Drew, 2022*). Abnormal expression of ORs has been observed in specific tumor cells and is closely associated with the development and progression of various cancers, including prostate, breast, and colorectal cancers (*Li et al., 2021*; *Morita et al., 2016*; *Pronin & Slepak, 2021*). Consequently, ORs may have a significant impact on the accurate diagnosis and targeted treatment of tumors. Recent studies have suggested a correlation between glioma formation and olfactory stimulation (*Chen et al., 2022*; *Kebir et al., 2020*). However, there are few studies on the role of ORs in gliomas, highlighting the need for further investigation into the underlying mechanisms and potential clinical implications of ORs in tumor biology.

In this study, we conducted, for the first time, a pan-cancer analysis of the olfactory receptor family 7 subfamily A member 5 (OR7A5) using public databases to elucidate its role in cancer. This comprehensive analysis encompasses expression patterns, prognostic correlations, DNA methylation, genetic and epigenetic features, and immune infiltration. Additionally, we used the Chinese Glioma Genome Atlas (CGGA) database to examine the correlation between OR7A5 expression and the clinicopathological characteristics of patients with glioma. Importantly, the functional role of OR7A5 in gliomas was validated through *in vitro* experiments, and its potential mechanisms of action were explored.

## MATERIALS & METHODS

### Expression analysis of OR7A5

We obtained mRNA expression data for OR7A5 in normal tissues from the Human Protein Atlas (HPA) project (http://www.proteinatlas.org) (*Uhlen et al., 2017*). The immunohistochemistry data for OR7A5 in normal and tumor tissues were sourced from the "Human Pathology" module of the HPA. Bulk tissue gene expression and single tissue expression of OR7A5 were retrieved from the Genotype-Tissue Expression (GTEx) database (http://www.gtexportal.org).

The expression levels of OR7A5 in both normal and cancer tissues were analyzed using the Tumor Immune Estimation Resource, version 2 (TIMER2) website

(http://timer.cistrome.org/) (*Li et al., 2020*). As the TIMER2 database did not encompass all cancer tissues and their corresponding normal tissue samples, additional analysis of testicular germ cell tumors (TGCT) was conducted using gene expression profiling interactive analysis, version 2 (GEPIA2) (http://gepia2.cancer-pku.cn/) (*Tang et al., 2019*). Expression analysis of OR7A5 in different tumor subtypes for head and neck squamous cell carcinoma (HNSC), lung squamous cell carcinoma (LUSC), and skin cutaneous melanoma (SKCM) was sourced from the tumor-immune system interaction database (TISIDB) web portal (http://cis.hku.hk/TISIDB) (*Ru et al., 2019*). The protein expression levels of OR7A5 were analyzed using the UALCAN website (http://ualcan.path.uab.edu) (*Chandrashekar et al., 2017*). Gene set enrichment analysis (GSEA) scores for OR7A5 were obtained using the Gene Set Cancer Analysis (GSCA) online website (http://bioinfo.life.hust.edu.cn/GSCA) (*Liu et al., 2018*), and the association between OR7A5 expression and signaling pathways was examined.

## Survival analysis

Associations between OR7A5 expression and overall survival across various cancer types were analyzed using TISIDB and GEPIA2. A Kaplan–Meier Plotter was employed to assess the correlation between OR7A5 expression levels and overall survival in different tumor types (http://kmplot.com/analysis/) (*Lánczky & Gyorffy, 2021*). GSCA was used to illustrate the survival differences between the high and low OR7A5 expression groups, including disease-free interval (DFI), disease-specific survival (DSS), overall survival (OS), and progression-free survival (PFS).

## Genetic and epigenetic alteration analysis

Genetic and epigenetic alterations in OR7A5 were analyzed using cBioPortal (http://www.cbioportal.org) (*Cerami et al., 2012*). The mutation frequency of OR7A5 was assessed across different tumors, along with an analysis of mutation counts in various cancer types. For patients harboring mutated or wild-type OR7A5, the impact of OR7A5 on disease-free survival (DFS) and PFS was investigated. Subsequently, detailed mutation types, such as missense, truncating, in-frame, splicing, and fusion mutations, were annotated. Furthermore, GSCA was used to analyze the percentages of single nucleotide variants (SNVs) and copy number variants (CNVs) in OR7A5. Additionally, GSCA explored survival differences between patients with cancer with mutated and wild-type OR7A5 as well as the correlation between different CNV events and overall tumor survival.

## Methylation of OR7A5 in cancers

Using GSCA, we analyzed the methylation differences in OR7A5 across various cancer types and explored the correlation between methylation and OR7A5 mRNA expression. Furthermore, GSCA was used to investigate the survival disparities (DFI, DSS, OS, and PFS) between patients with cancer with high and low levels of OR7A5 methylation levels.

## Tumor-immune system and OR7A5

Using the TISIDB dataset, we analyzed the relationship between tumor-infiltrating lymphocytes (TILs), immunoinhibitors, immunostimulators, major histocompatibility
complex (MHC) molecules, chemokines, receptors, and OR7A5. Furthermore, TISIDB was used to assess whether there were significant expression or mutation differences of OR7A5 between responders and non-responders to immunotherapy. Sangerbox 3.0 (http://www.sangerbox.com/home.html) was used to investigate the association of OR7A5 with microsatellite instability (MSI) and tumor mutational burden (TMB).

## Immune infiltration in cancer

We used TIMER2 to assess the correlation between OR7A5 expression and immune infiltration across various cancer types, including B cells, CD4$^+$ T cells, CD8$^+$ T cells, dendritic cells, macrophages, monocytes, NK cells, neutrophils, T cell regulatory (Tregs), mast cells, cancer-associated fibroblasts (CAFs), endothelial cells, eosinophils, hematopoietic stem cells (HSCs), common lymphoid progenitor, common myeloid progenitor, granulocyte-monocyte progenitor, T cell NK, and myeloid-derived suppressor cells (MDSCs).

## Analysis of OR7A5 expression and with clinicopathological characteristics and prognosis

Using expression data and survival information obtained from TCGA database for patients with glioma, we analyzed the expression levels of OR7A5 in gliomas and its association with OS and DFS. Additionally, using the CGGA database (http://www.cgga.org.cn/), we investigated variations in OR7A5 mRNA expression levels among different clinical factors in patients with glioma, encompassing sex, age, grade, recurrence status, IDH mutation status, and 1p/19q co-deletion status.

## Analysis of pathway correlation

Using the glioma RNA-seq data acquired from TCGA database, the genes encompassing the respective pathways were gathered and subsequently analyzed using the R software GSVA package. The parameter method ='ssgsea' was selected, followed by an examination of the correlation between genes and pathway scores *via* Spearman correlation analysis (*Xiao, Dai & Locasale, 2019*).

## Cell culture

The U87 and U251 human glioma cell lines were obtained from the National Collection of Authenticated Cell Cultures (Shanghai, China). These cells were cultured in Dulbecco's Modified Eagle Medium supplemented with 10% fetal bovine serum from Gibco, USA and 1% penicillin (100 U/mL; BI, Cromwell, CT, USA)-streptomycin (100 µg/mL). Cell cultures were incubated at 37 °C and 5% $CO_2$.

## siRNA transfection

In accordance with the manufacturer's instructions, siRNA transfection was performed using Lipofectamine TM 2000 reagent (Invitrogen, Waltham, MA, USA). OR7A5-targeted and control siRNAs were procured from General Biosystems (Anhui, China). The sequences for si-RNA-1 were 5′-AGAACAAAGUCAUCACCUA-3′, for si-RNA-2 were 5′-ACAAGAACCUGGACUGCAA-3′, and for si-RNA-3 were 5′-GCUACAAAUCUUAAUGGUA-3′.

## Cell viability assay

Transfected cells were transferred to a 96-well plate at a density of $2 \times 10^3$ cells/well. Subsequently, the plate should be incubated in a carbon dioxide incubator at a temperature of 37 °C. At 24, 48, 72, and 96 h of cultivation, 10 µL of CCK-8 solution (Beyotime Biotechnology, Shanghai, China) should be added to each well and incubated for 1 h. The absorbance was measured at 450 nm.

## Colony formation assay

Transfected cells were seeded at a density of $2 \times 10^3$ cells/well in six-well plates and incubated for 14 days. After rinsing with phosphate-buffered saline, cells were immobilized with 4% paraformaldehyde for 15 min and stained with Giemsa staining solution (Solarbio, Beijing, China).

## EdU-incorporation experiments

Cultured cells were seeded at a density of $3 \times 10^4$ cells/well in six-well plates and allowed to adhere. Subsequently, a diluted EdU solution was added according to the instructions provided with the EdU Assay Kit (Sangon, Shanghai, China). The cells were then incubated in a $CO_2$ incubator for 2 h, fixed using 4% paraformaldehyde, and stained with the Hoechst staining solution. Finally, the cells were sealed and images were captured.

## Determination of triglyceride and total cholesterol levels

According to the manufacturer's instructions, triglyceride and total cholesterol levels were measured using the Triglyceride Assay Kit (Nanjing Jiancheng Bioengineering Institute, Nanjing, Jiangsu, China) and the Total Cholesterol Assay Kit (Nanjing Jiancheng Bioengineering Institute, Nanjing, Jiangsu, China). A suspension of $1 \times 10^6$ cells was prepared and sonicated, and 2.5 µL was added to a 96-well plate along with an equivalent volume of distilled water and standard samples. Subsequently, 250 µL of working solution was added to each well, and the plate was incubated at 37 °C for 10 min. Absorption at 510 nm was measured using a full-wavelength spectrophotometer.

## Western blotting

Initially, proteins were extracted using a highly efficient lysis buffer containing PMSF (Solarbio, Beijing, China). Subsequently, proteins were separated using 8% and 10% SDS-PAGE, followed by transfer onto polyvinylidene fluoride (PVDF) membranes (Millipore). The PVDF membranes were then incubated overnight at 4 °C with primary antibodies (OR7A5 from HuaAnBio, Hangzhou, China; FASN, SREBP1, β-actin, and GAPDH obtained from Santa Cruz Biotechnology, Inc.). The following day, the corresponding secondary antibodies (mouse secondary antibody at 1:5000 and rabbit secondary antibody at 1:5000, purchased from Zhongshan Golden Bridge Biotechnology Co., Ltd.) were applied and incubated at room temperature for 1.5 h. Protein expression levels were visualized by adding ECL chemiluminescence liquid (Solarbio, Beijing, China) and observed using an electronic imaging system. Finally, quantitative analysis was performed using ImageJ software.

## Statistical analysis

Statistical analyses were conducted using SPSS 26.0 and GraphPad Prism 8.0. Group comparisons of the quantitative data were performed using $t$-tests. All results were validated through three independent experiments, and the values are presented as mean $\pm$ standard deviation. A significance level of $p < 0.05$ was considered indicative of statistical significance.

# RESULTS

## Aberrantly expressed ORs in gliomas

During the preliminary stages of our research, we conducted thorough literature reviews, revealing that ORs are intricately linked to the occurrence and progression of various malignancies, such as prostate cancer, breast cancer, and colorectal cancer. However, the precise role of ORs in gliomas remains elusive, prompting us to delve into investigating their functionality in this context. To evaluate the expression levels of ORs in gliomas, we accessed RNA-seq data and associated clinical information encompassing 666 samples of lower-grade glioma (LGG) and glioblastoma (GBM) tumors from the TCGA database. Our analysis identified ten ORs exhibiting significant differential expression: OR52N4, OR2L13, OR7D2, OR4N2, OR2C1, OR3A2, OR7A5, OR13A1, OR2W3, and OR2A9P (Fig. S1A). To determine the correlation between these genes and glioma prognosis, we employed forest plots. Our findings indicated that the expression of eight genes was significantly associated with prognosis, while two genes showed no apparent correlation (Fig. S1B). Specifically, OR2C1, OR2L13, OR4N2, OR7D2, OR13A1, and OR52N4 displayed a negative correlation with unfavorable prognosis (Figs. S1C–S1H), whereas OR7A5 and OR2A9P exhibited a positive correlation (Figs. S1I and S1J). OR2A9P, a pseudogene of the olfactory receptor family 2 subfamily A member 9, was excluded from further analysis because of its lack of defined biological functions. Our initial analysis highlighted the noteworthy overexpression of OR7A5 in gliomas and its potential association with a poor prognosis in glioma patients. Given the limited research on OR7A5, we conducted an in-depth, pan-cancer analysis of its expression levels, genetic characteristics, and immune infiltration.

## OR7A5 expression levels in normal and tumor tissues

A flow-chart of the study is shown in Fig. 1. To assess the expression levels of OR7A5 in human normal tissues, we used the GTEx database to evaluate the expression profile of OR7A5 across various tissues. The results indicated that OR7A5 exhibited a certain degree of specificity in its expression in normal tissues, with elevated expression in the pituitary glands and testes (Fig. 2A). Subsequently, the OR7A5 expression levels in individual tissues were analyzed using the GTEx database (Fig. 2B). Furthermore, the HPA database revealed that OR7A5 was predominantly expressed at low levels in most normal tissues, with moderate expression observed in breast cancer, lung cancer, glioma, and melanoma tissues (Fig. 2C).

To detect the expression levels of OR7A5 in tumor tissues, we used TIMER2 to evaluate the differential expression of OR7A5 in tumor tissues and their corresponding adjacent normal tissues. The analysis revealed a significant upregulation of OR7A5 in bladder

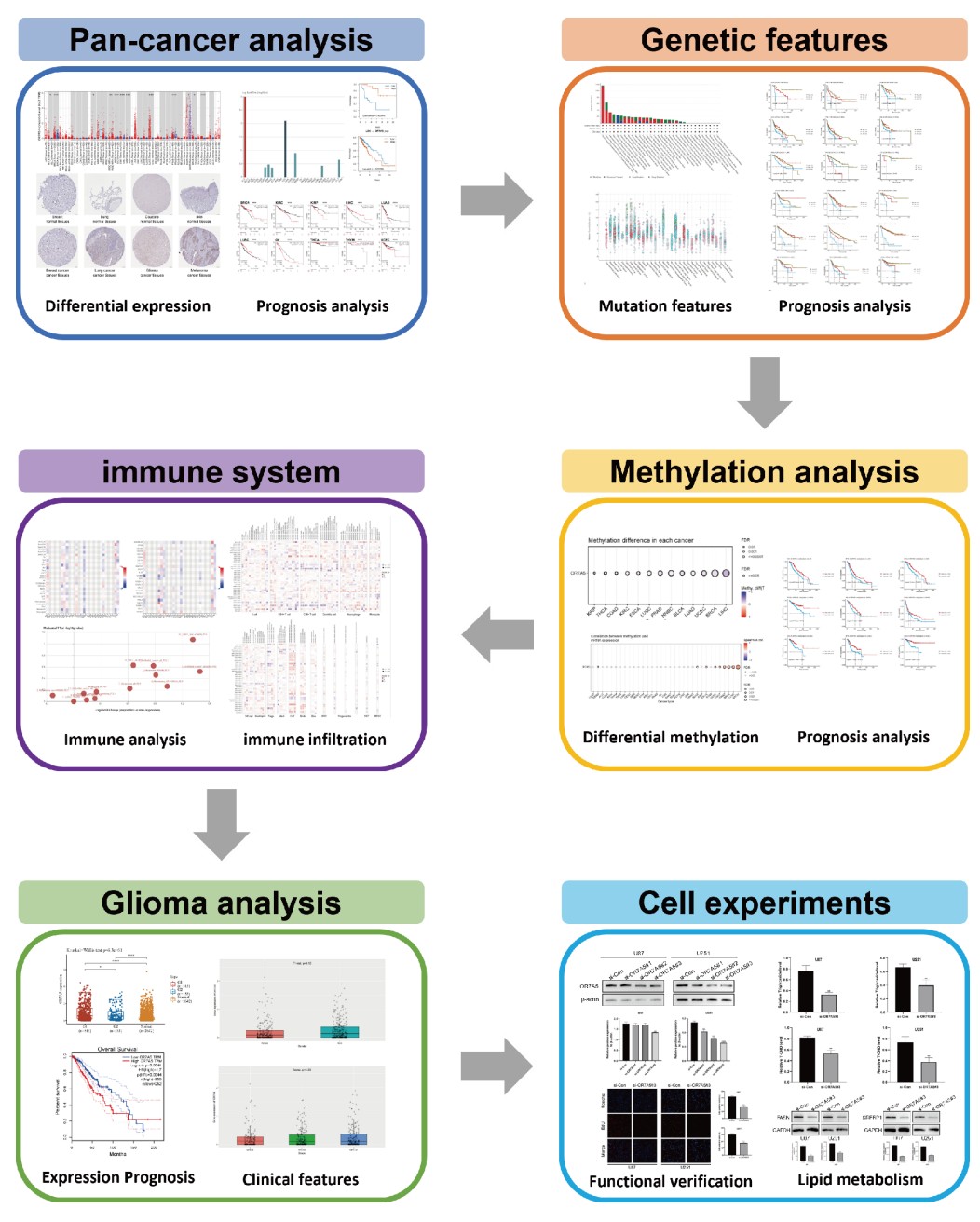

**Figure 1  The flow chart of this study.**

urothelial carcinoma (BLCA), breast invasive carcinoma (BRCA), esophageal carcinoma (ESCA), HNSC, kidney renal clear cell carcinoma (KIRC), kidney renal papillary cell carcinoma (KIRP), LUSC, SKCM, stomach adenocarcinoma (STAD), and thyroid carcinoma (THCA) (Fig. 3A). Due to the limited representation of OR7A5 expression data in cancerous and adjacent tissues in the TIMER2 database, notably lacking information on acute myeloid leukemia (AML), lower-grade glioma (LGG), mesothelioma (MESO), and testicular germ cell tumors (TGCT) among other malignancies, we opted to augment our

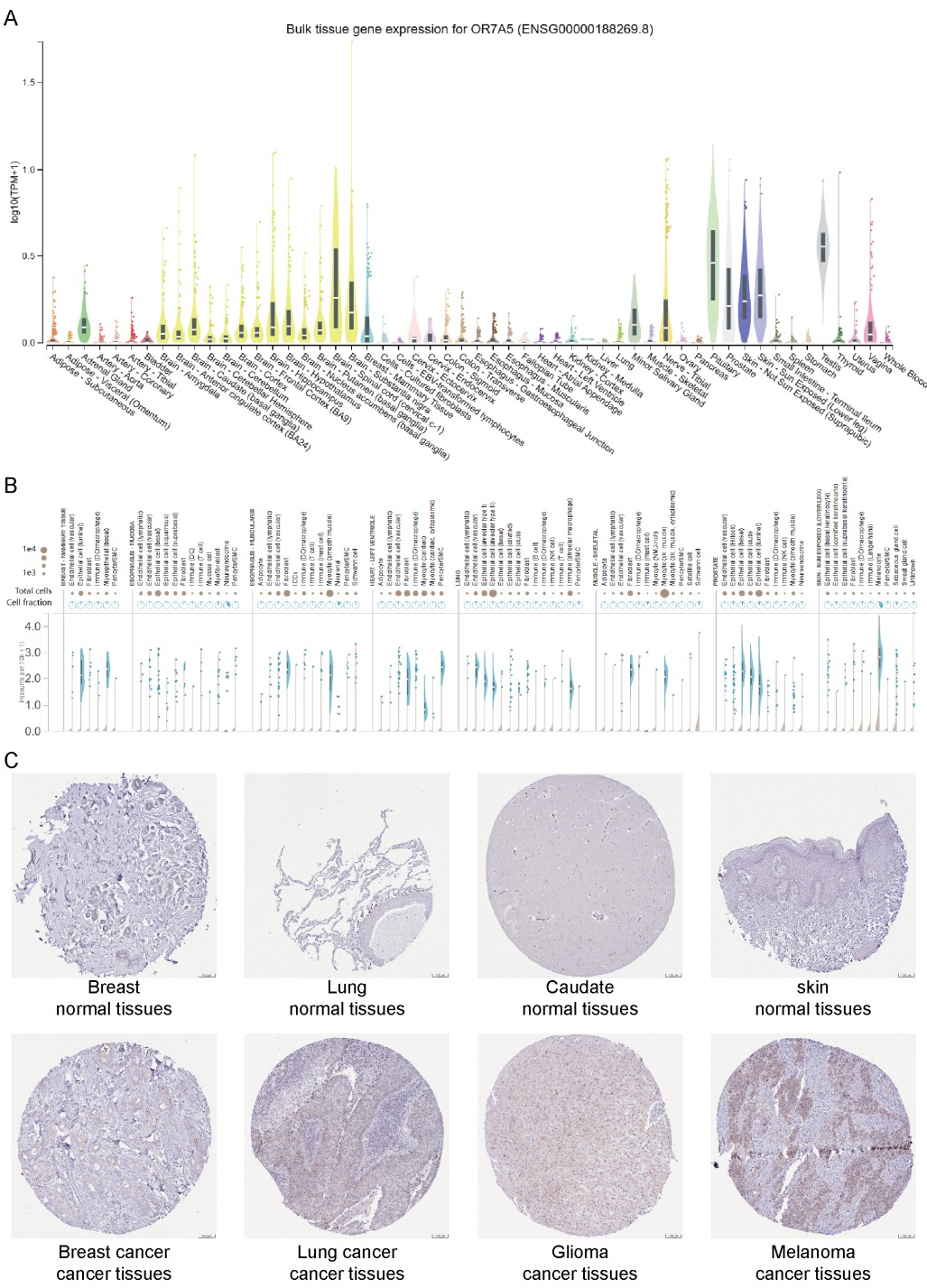

**Figure 2  Expression levels of OR7A5 in normal and cancer tissues.** (A) Analysis of OR7A5 expression in normal tissues through the GTEx database. (B) Expression levels of OR7A5 in single-cell analysis within individual tissues from the GTEx database. (C) IHC images of OR7A5 observed in normal tissues (breast, lung, caudate, and skin) and tumor tissues (breast cancer, lung cancer, glioma, and melanoma) from the HPA database.

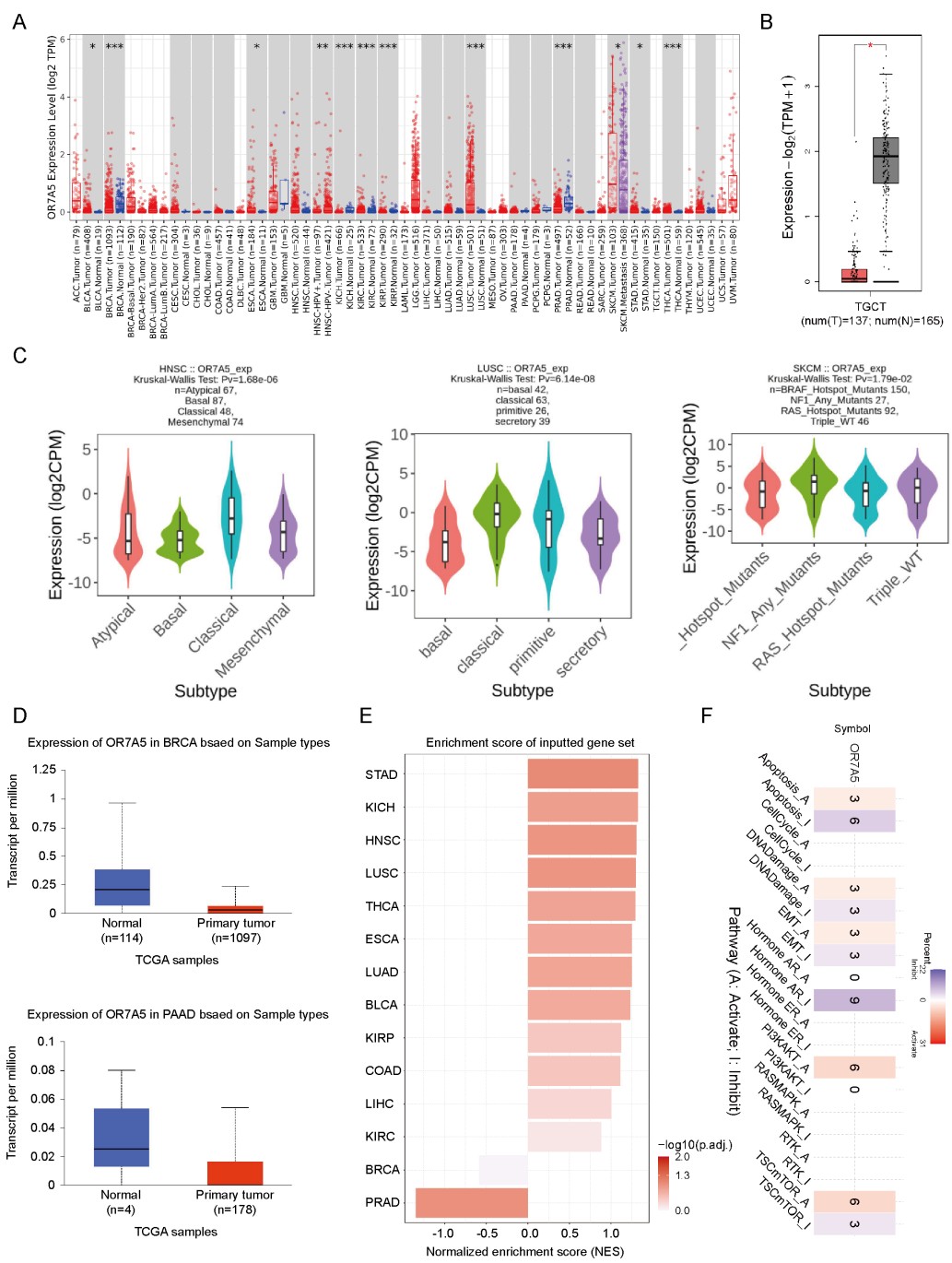

**Figure 3** **Expression levels of OR7A5 in different types of cancers.** (A) Analysis of the differential expression of OR7A5 in various cancer types and their corresponding adjacent tissues through TIMER2. (B) Expression levels of OR7A5 in testicular germ cell tumors (TGCT) analyzed using GEPIA. (C) Analysis of the expression status of OR7A5 in subtypes of head and neck squamous cell carcinoma (HNSC), lung squamous cell carcinoma (LUSC), and skin cutaneous melanoma (SKCM) through TISIDB. (D) Expression levels of OR7A5 in breast invasive carcinoma (BRCA) and pancreatic adenocarcinoma (PAAD) analyzed using GEPIA. (E) Enrichment scores of OR7A5 in different cancers analyzed through GSCA. (F) Analysis of signaling pathways associated with OR7A5 expression using GSCA.

analysis with the GEPIA2 database. We observed a significant decrease in OR7A5 expression levels in TGCT compared to that in adjacent tissues (Fig. 3B). Recognizing that distinct tumor subtypes have crucial implications for prognosis and therapeutic approaches, we delved deeper into the expression patterns of OR7A5 across various tumor subtypes utilizing the TISIDB database (Fig. 3C). For instance, HNSC is classified into four distinct molecular subtypes: atypical, basal, classical, and mesenchymal. Our analysis revealed a higher expression of OR7A5 in the classical subtype and lower expression in the basal subtype. Similarly, LUSC encompasses several subtypes, including basal, classical, primitive, and secretory, with a higher expression observed in the classical subtype. Cutaneous melanoma, whose etiology is associated with UV radiation, excessive sun exposure, and mutations in genes such as BRAF, NF1, and RAS, also showed slight differences in OR7A5 expression when analyzed in the TISIDB database. Specifically, we observed a higher expression of OR7A5 in the NF1 mutation subtype compared to the wild-type subtype. To broaden our understanding of OR7A5 expression in diverse tumors, we utilized multiple databases. Analysis through the UALCAN database confirmed a significantly lower expression of OR7A5 in BRCA and pancreatic adenocarcinoma (PAAD) tissues compared to normal tissues, aligning with our previous findings depicted in Fig. 3A (Fig. 3D). Furthermore, our exploration of the GSCA database revealed insightful enrichment scores of OR7A5 across diverse cancer types. The analysis indicated significant OR7A5 expression in 14 tumors, including STAD, kidney chromophobe (KICH), and HNSC, whereas its expression was comparatively lower in BRCA and prostate adenocarcinoma (PRAD) tumors (Fig. 3E). In addition, we investigated the pathways influenced by OR7A5 mRNA expression using the GSCA database. Notably, Fig. 3F illustrates significantly inhibited pathways such as apoptosis and hormone AR pathways, whereas significantly activated pathways included the PI3K/AKT and TSC/mTOR signaling pathways.

## Prognostic significances of OR7A5 in different cancers

OR7A5 expression is elevated in various tumors. We therefore explored the prognostic relevance of OR7A5 in cancer cells. Initially, analysis based on TISIDB revealed that high OR7A5 expression was associated with shorter OS in LGG and longer OS in adrenocortical carcinoma (ACC) (Fig. 4A). Further individualized analyses revealed that, in ACC, patients exhibiting high OR7A5 expression enjoyed significantly longer OS compared to those with low expression. However, in LGG, the converse was true, with patients having higher OR7A5 expression experiencing significantly shorter OS (Fig. 4B). Given the limited prognostic outcomes available in the TISIDB, we broadened our search to other databases. GEPIA2 allowed us to analyze RNA sequencing data encompassing 9736 tumors and 8587 normal samples from TCGA and GTEx. This analysis revealed that BRCA patients with high OR7A5 expression exhibited longer OS compared to those with low expression. Additionally, we observed an association between high OR7A5 expression and poor prognosis in patients with LGG, which was consistent with results from the TISIDB database (Fig. 4C). However, the prognostic insights from GEPIA2 had their limitations. Therefore, we delved deeper into the data using the Kaplan–Meier Plotter, a widely used tool for survival analysis. This platform, which aggregates data from GEO, EGA, and TCGA databases, enables us to

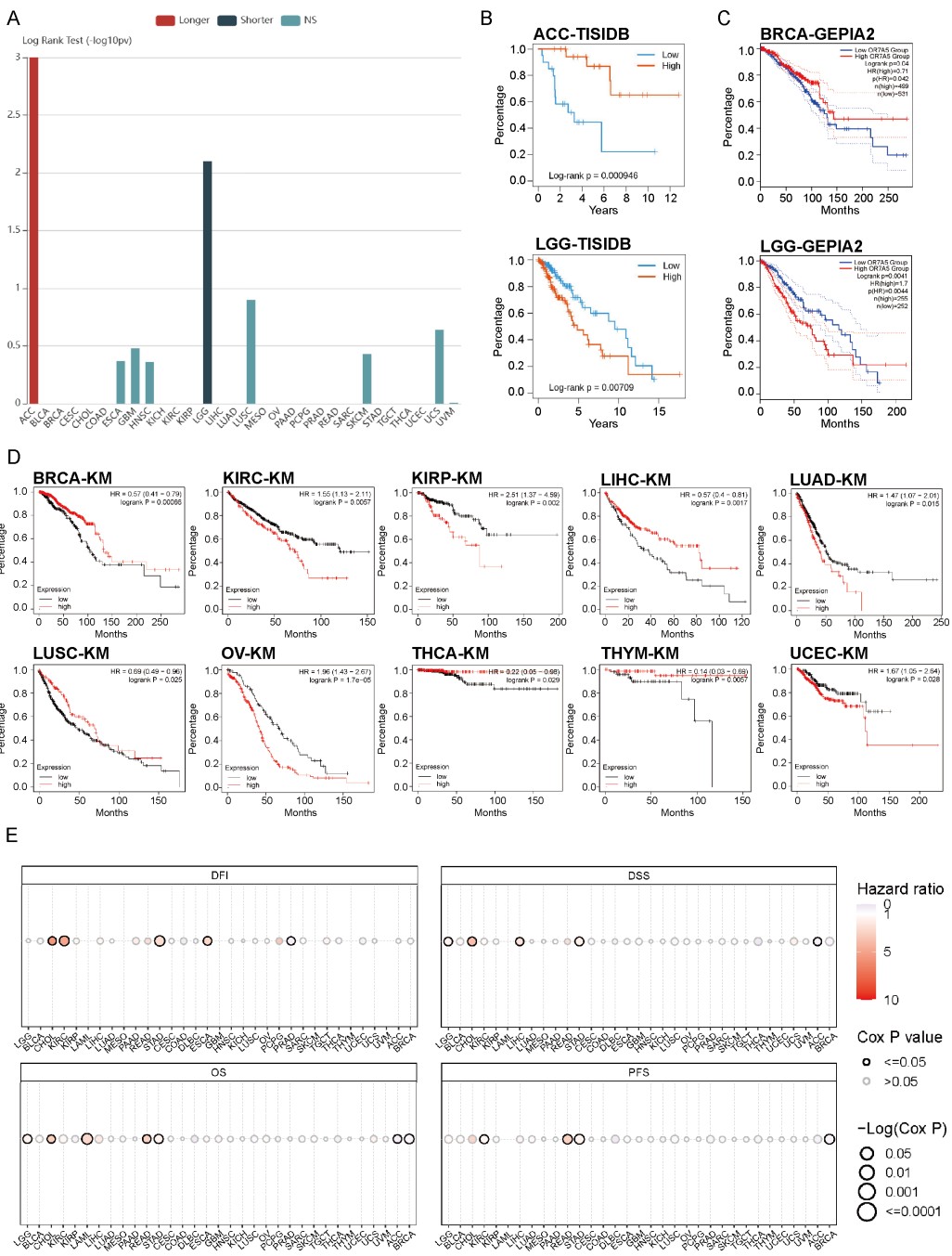

**Figure 4 Correlation analysis between OR7A5 expression and cancer prognosis.** (A) Analysis of the association between OR7A5 expression levels and the prognosis of various cancers through TISIDB. (B) Analysis of the correlation between OR7A5 expression and the prognosis of adrenocortical carcinoma (ACC) and lower-grade glioma (LGG) using TISIDB. (C) Examination of the relationship between OR7A5 expression and the prognosis of breast invasive carcinoma (BRCA) and LGG through GEPIA. (D) Kaplan–Meier analysis demonstrating the impact of OR7A5 expression levels on the overall survival of various cancers. (E) Observation of differences in survival periods between high and low OR7A5 expression groups through GSCA.

assess the correlation between gene expression and patient survival rates across over 30,000 samples of 21 tumor types. As depicted in Fig. 4D, patients with high OR7A5 expression exhibited better survival rates in BRCA, liver hepatocellular carcinoma (LIHC), LUSC, THCA, and thymoma (THYM), whereas patients with KIRC, KIRP, lung adenocarcinoma (LUAD), ovarian serous cystadenocarcinoma (OV), and uterine corpus endometrial carcinoma (UCEC) demonstrated lower survival rates. All these analyses focused on OS, but metrics such as DFI, DSS, and PFS are equally crucial. Consequently, we utilized the GSCA database to conduct a more comprehensive analysis of these metrics. The analysis revealed an association between OR7A5 and DFI in cholangiocarcinoma (CHOL), KIRC, STAD, ESCA, and PRAD; DSS in LGG, CHOL, LIHC, STAD, and ACC; OS in LGG, CHOL, LAML, rectum adenocarcinoma (READ), STAD, ACC, and BRCA; and PFS in KIRC, READ, STAD, and BRCA (Fig. 4E).

## Genetic alterations of OR7A5 in human cancers

In this study, we delved into the genetic mutation profile of OR7A5 across a broad range of human cancers. Using the cBioPortal platform, we identified a total of 60 mutations pertaining to OR7A5 (Fig. 5A). The highest frequency of OR7A5 alterations (>10%) was observed in OV, with amplification being the predominant mutation type (Fig. 5B). Further analysis revealed the general mutation counts of OR7A5 across different cancer types. As shown in Fig. 5C, the mutation types mainly included shallow deletions, gains, and amplifications. Notably, OR7A5 mutations were found to impact patient prognosis, with a significantly lower PFS and DFS observed in patient with cancer and OR7A5 mutations than in those without mutations (Fig. 5D). The frequency of OR7A5's SNVs and CNVs varied considerably across tumors. Specifically, UCEC exhibited the highest SNV mutation frequency (Fig. 5E). Moreover, significant differences in survival (DFI, DSS, OS, and PFS) were observed between the SNV group and wild-type (WT) groups. Particularly, patients with BLCA harboring OR7A5 mutations displayed a poorer PFS (Fig. S2A). The CNV percentage of OR7A5 across various tumors is depicted in Fig. 5F, and noteworthy survival differences were observed between the different CNV groups (Fig. S2B). This comprehensive analysis provides valuable insights into the genetic landscape of OR7A5 in cancer and its potential implications for patient prognosis.

## OR7A5 methylation in cancers

DNA methylation stands as a pivotal factor in regulating cancer progression. The GSCA database was used to explore the methylation status of OR7A5 in various cancers. Our analysis revealed significant differences in methylation status between tumor and normal samples in KIRP, THCA, colon adenocarcinoma (COAD), KIRC, ESCA, LUSC, PRAD, HNSC, BLCA, LUAD, UCEC, BRCA, and LIHC (Fig. 6A). Given the critical role of DNA methylation in gene expression regulation, we sought to examine the link between OR7A5's mRNA expression levels and its methylation status. We uncovered a negative correlation between OR7A5's DNA methylation and its expression in rectum adenocarcinoma (READ), whereas a positive correlation was observed in BLCA, ESCA, UCEC, TGCT, PRAD, BRCA, LUSC, and SKCM (Fig. 6B). To further explore the potential clinical significance of
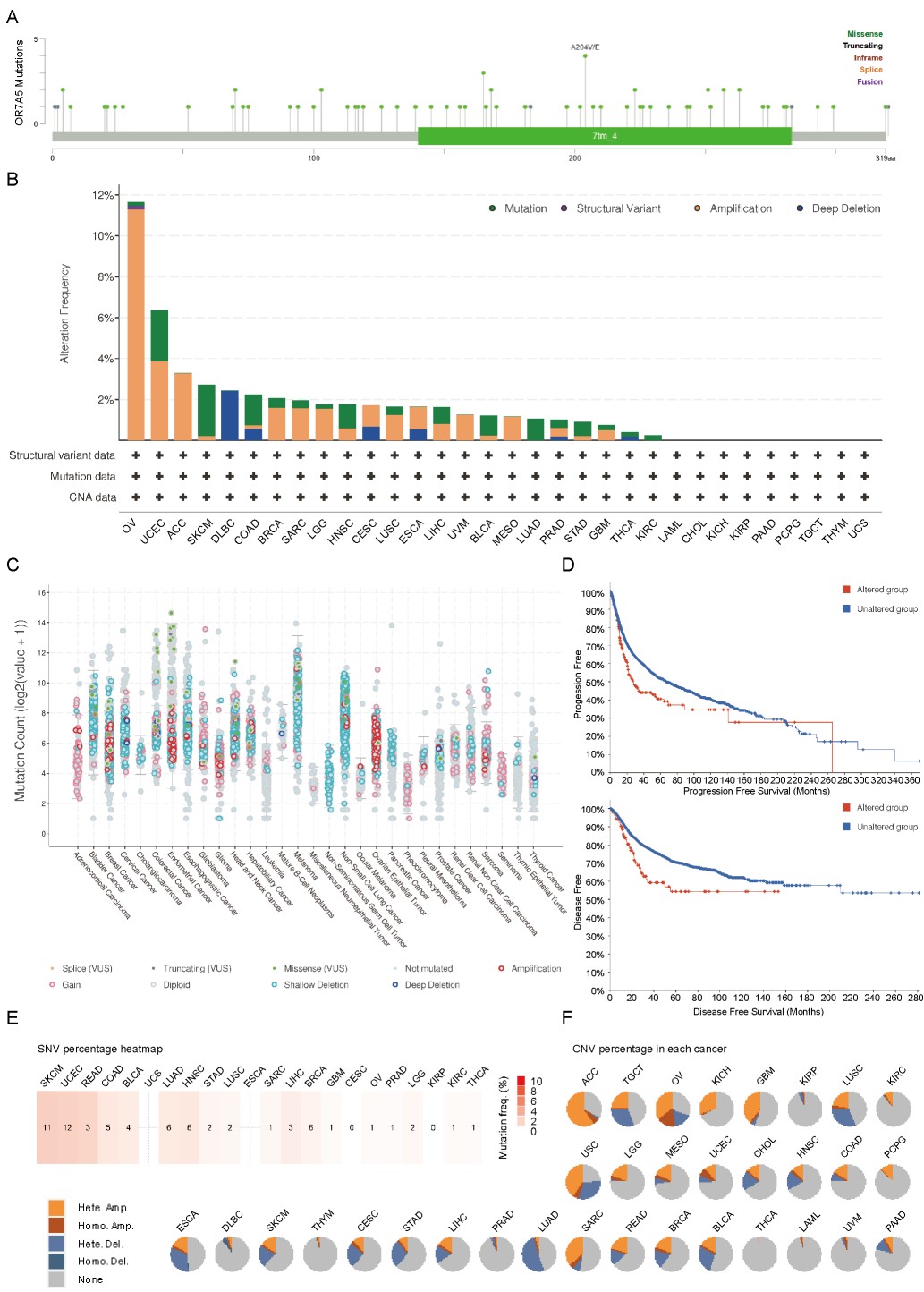

**Figure 5  Genetic characteristics of OR7A5 in different tumors.** (A) Analysis of mutation types and sites of OR7A5 through cBioPortal. (B) Alteration frequencies of OR7A5 in different tumors. (C) Analysis of OR7A5 mutation counts through cBioPortal. (D) Impact of the mutation status of OR7A5 on the survival period of patients with cancer. (E) Analysis of OR7A5 single nucleotide variants (SNVs) in cancers through a heatmap. (F) Analysis of OR7A5 copy number variation (CNV) in each cancer type through GSCA.

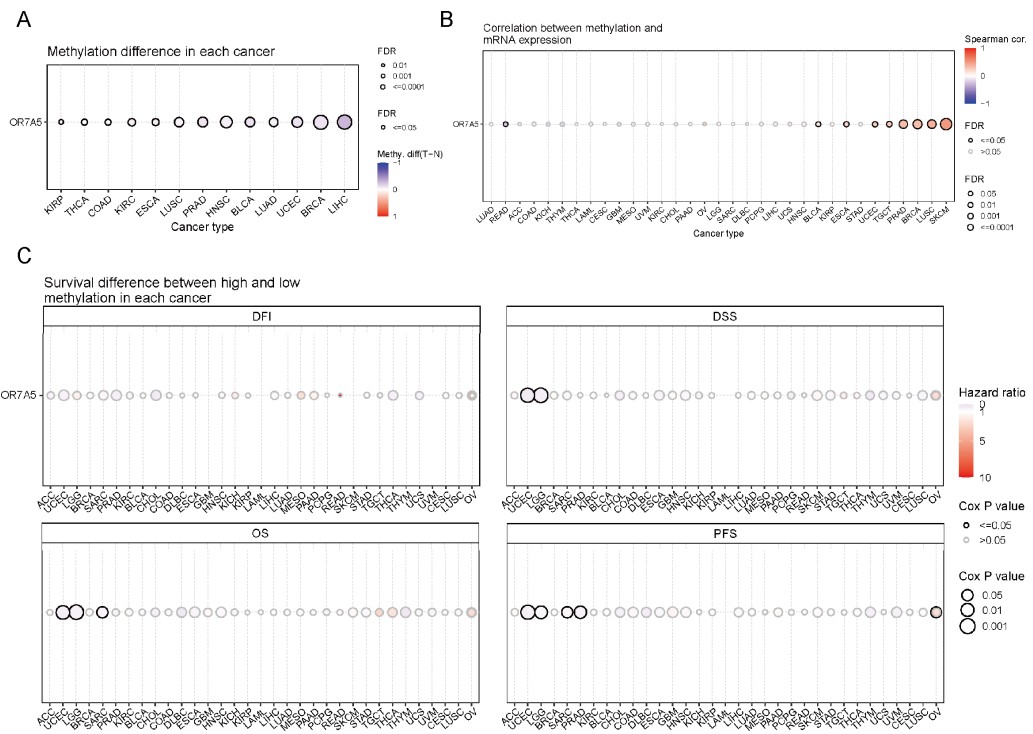

**Figure 6** **Methylation analysis of OR7A5 in different cancer types.** (A) Analysis of the methylation differences of OR7A5 in various tumors through GSCA. (B) Assessment of the correlation between OR7A5 mRNA expression and methylation through GSCA. (C) Examination of the survival differences between high and low methylation of OR7A5 through GSCA.

OR7A5's methylation levels, we assessed their impact on patient prognosis. The GSCA database provides four survival indicators: DFI, DSS, OS, and PFS. While DFI serves as an important clinical metric, its recording poses challenges due to the need for continuous disease recurrence monitoring. DSS reflects the clinical benefits specific to a disease, while OS remains a widely used survival metric. PFS, though typically based on a smaller sample size and shorter follow-up, offers higher reliability by being less affected by crossover and subsequent treatments. Our survival analysis indicated that the methylation level of OR7A5 was not associated with DFI in various tumor types. However, high methylation levels of OR7A5 correlated with favorable DSS in UCEC and LGG, favorable OS in UCEC, LGG, and sarcoma (SARC), and improved PFS in UCEC, LGG, SARC, and PRAD (Fig. 6C). These findings provide valuable insights into the potential role of OR7A5 methylation in cancer prognosis.

## Correlation between OR7A5 expression and immune cell infiltration

The infiltration of immune cells holds a pivotal role in cancer progression, and to delve deeper into this phenomenon, we employed the TIMER2 database. Through various immune infiltration methods, we investigated the interplay between OR7A5 expression and the infiltration of immune cells across diverse cancer types. Our analysis uncovered a significant correlation between OR7A5 expression and the abundance of various immune

cell types across multiple cancer types. This included a notable association with B cells in 34 cancer types, CD4$^+$ T cells in 38 cancer types, CD8$^+$ T cells in 28 cancer types, myeloid dendritic cells in 29 cancer types, macrophages in 38 cancer types, monocytes in 28 cancer types, NK cells in 28 cancer types, neutrophils in 24 cancer types, regulatory T cells in 22 cancer types, mast cells in 17 cancer types, CAFs in 26 cancer types, endothelial cells in 24 cancer types, eosinophils in 11 cancer types, HSCs in 13 cancer types, progenitor cells in 12 cancer types, NKT cells in 14 cancer types, and MDSCs in 13 cancer types (Fig. 7). These findings offer valuable insights into the complex interplay between OR7A5 expression and immune cell infiltration in cancer.

## Relationship between OR7A5 and the immune system

With a focused aim to gain a deeper understanding of the intricate relationship between OR7A5 and the tumor microenvironment, we harnessed the TISDB database to explore potential associations between OR7A5 and various immunological factors. These included TILs, immune inhibitors, immune stimulators, MHCs, chemokines, and receptors (Figs. 8A–8F). The results revealed significant correlations between OR7A5 expression and various immune modulators in ACC, ESCA, GBM, HNSC, LGG, LUSC, SKCM, uterine carcinosarcoma (UCS), and uveal melanoma (UVM). These findings suggest that OR7A5 plays an indispensable role in modulating immune responses in human cancers.

## Correlation between OR7A5 expression and TMB and MSI

The TMB an emerging biomarker, has been found to correlate with the efficacy of immune checkpoint inhibitors in a large number of clinical studies, and its role in predicting the efficacy of tumor immunotherapy is increasingly recognized in clinical practice. In our analysis, we observed a correlation between OR7A5 expression and TMB in human cancer cells. Notably, OR7A5 expression displayed a significant positive correlation with TMB in SKCM and LUAD, while exhibiting a significant negative correlation in ESCA, UCS, KIRP, and LUSC (Fig. 9A). MSI is closely associated with tumorigenesis and is caused by defects in mismatch repair (MMR) genes. MSI or MMR testing holds immense clinical value in diagnosis, prognosis, and treatment selection for various solid tumors. Here, we delved into the relationship between OR7A5 expression and MSI in human cancers. OR7A5 expression showed a significant positive correlation with MSI in UVM and SKCM, whereas it displayed a negative correlation in diffuse large B-cell lymphoma (DLBC), CHOL, KICH, LAML, ESCA, PRAD, UCS, and MESO (Fig. 9B). Collectively, these findings indicate that OR7A5 impacts anti-tumor immunity by modulating the composition and immunological mechanisms within the tumor microenvironment.

## Correlation between OR7A5 expression levels and the survival and clinical characteristics of patients with glioma

The expression levels of OR7A5 in glioma and normal tissues were analyzed using TCGA and GTEx databases. These findings demonstrated a significant upregulation of OR7A5 expression in both LGG and GBM compared to normal tissues (Fig. 10A). Survival analysis indicated that patients with high OR7A5 expression had notably lower OS and DFS than those with low OR7A5 expression (Figs. 10B and 10C). The CGGA dataset was used to

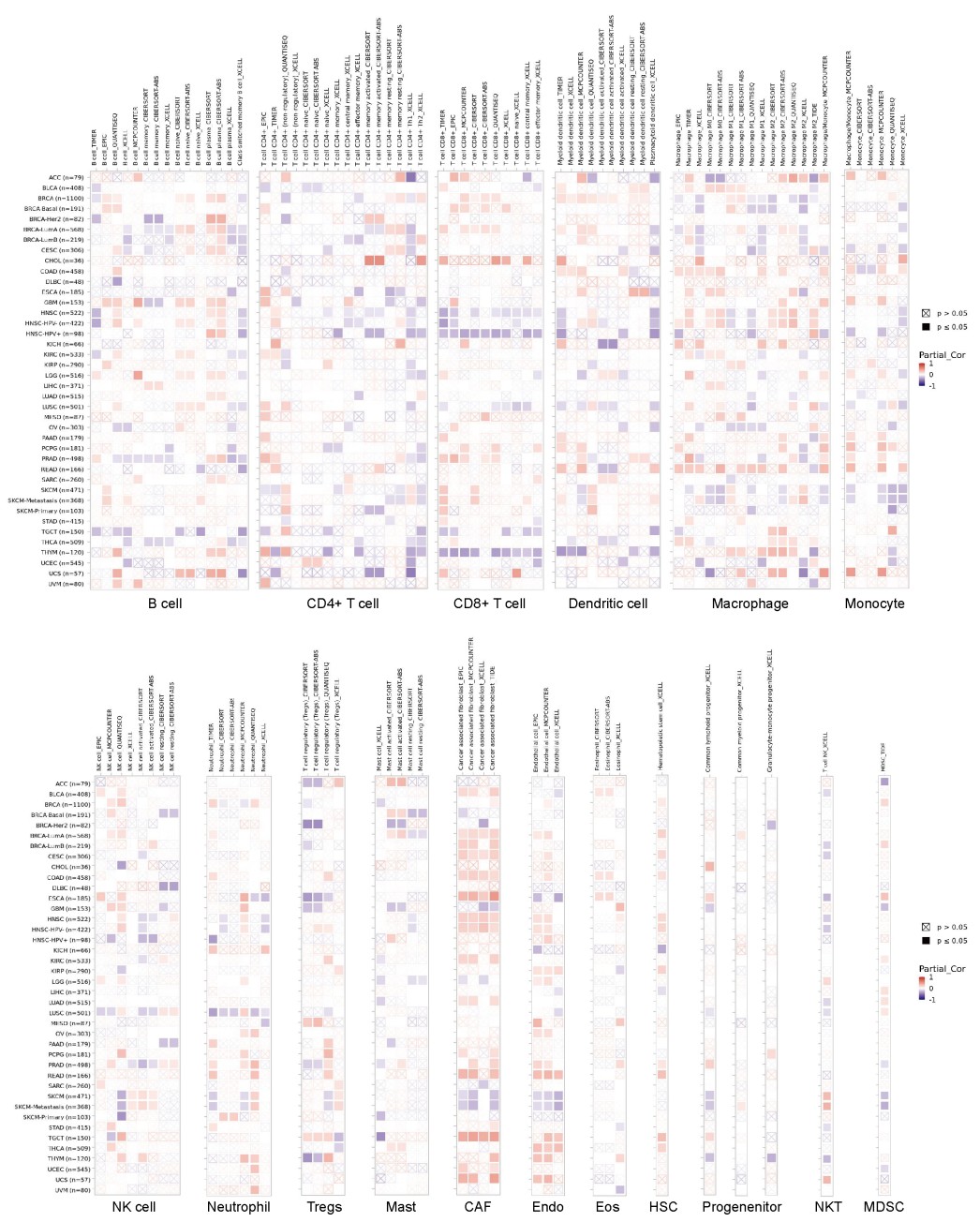

**Figure 7** Analysis of the correlation between OR7A5 expression and immune infiltration through TIMER2.

further investigate the relationship between OR7A5 expression and clinicopathological features. The results showed no significant correlation between the expression level of OR7A5 and sex, age, grade, or disease progression (Figs. 10D–10G). Notably, a significant difference in OR7A5 expression was observed between the 1p/19p co-deletion and non-co-deletion states in the CGGA dataset, with higher expression in the non-co-deletion state (Fig. 10H). Additionally, a significant difference in OR7A5 expression was found between

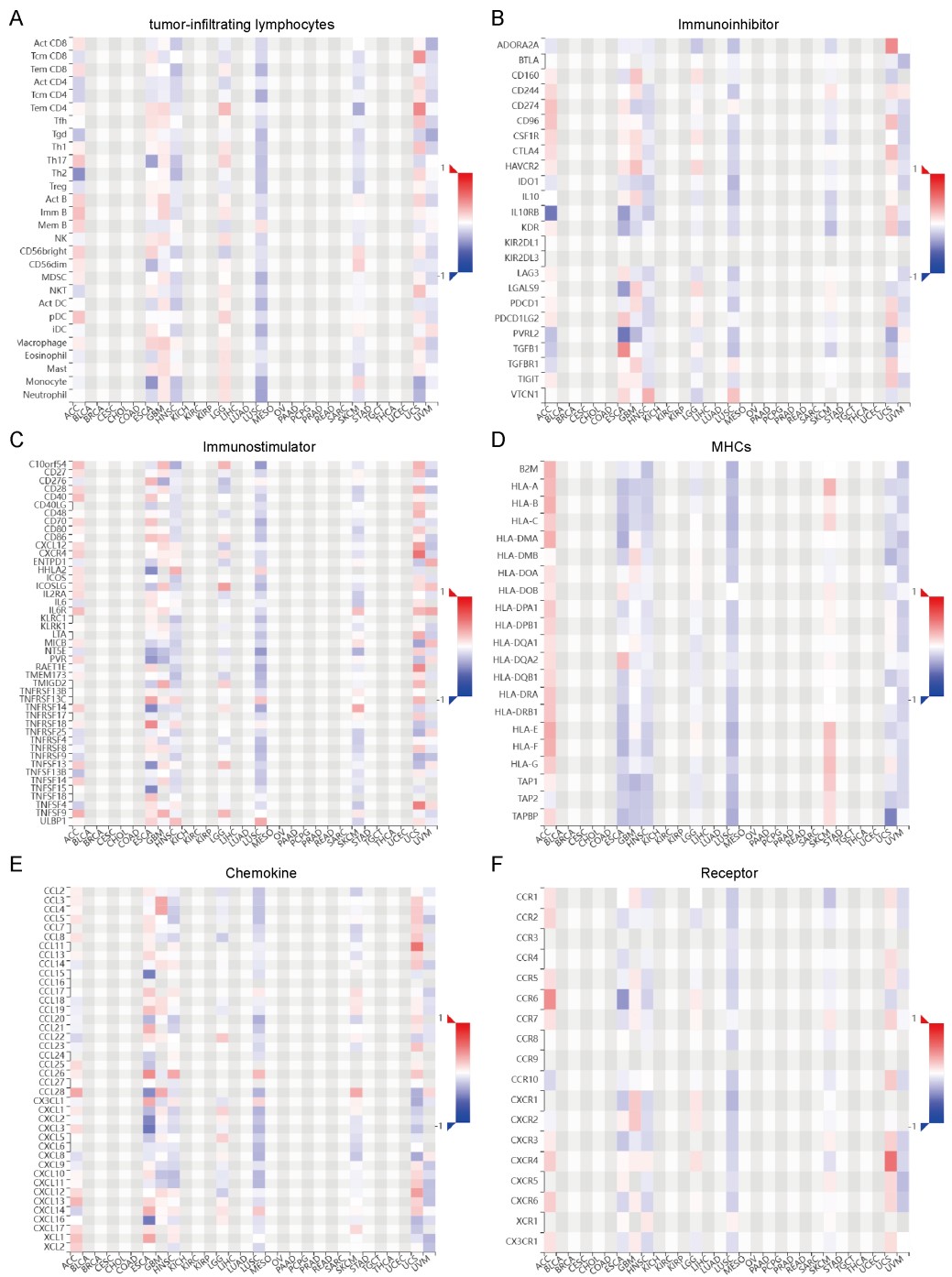

**Figure 8** **Relationship between OR7A5 and the tumor immune system.** Analysis of the correlation between OR7A5 expression and TILs (A), immunoinhibitors (B), immunostimulators (C), MHC (D), chemokines (E), and receptors (F) through TISIDB.

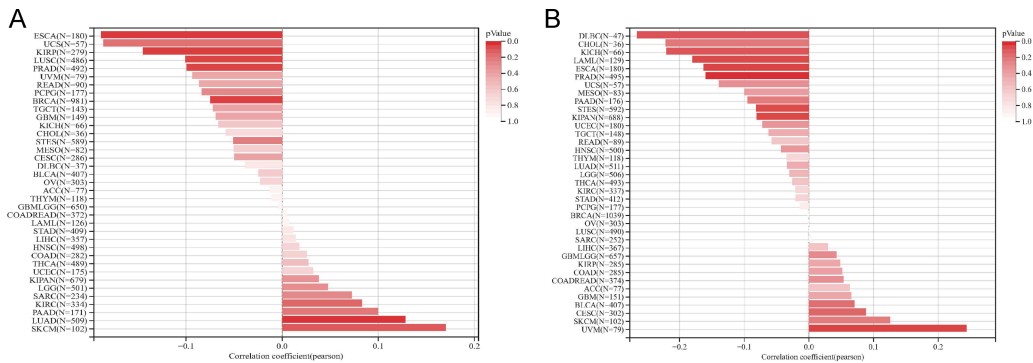

**Figure 9** Assessment of the association between OR7A5 expression in different cancers and the TMB (A) and MSI (B) through the Sangerbox online platform.

IDH-mutant and IDH-wild-type cases, with higher expression in the IDH-wild-type group (Fig. 10I).

## OR7A5 enhanced the proliferative capacity of glioma cells

To investigate the role of OR7A5 in gliomas, three distinct siRNA-OR7A5 sequences were transfected into U87 and U251 glioma cells to establish cell lines with reduced OR7A5 expression. Western blot analysis revealed a significant decrease in OR7A5 expression levels in the si-OR7A5#2 and si-OR7A5#3 groups compared to the control group (si-Con), with the si-OR7A5#3 group exhibiting the most pronounced silencing effect. Consequently, the cells derived from the si-OR7A5#3 group were selected for subsequent experiments (Fig. 11A). Subsequently, to examine the involvement of OR7A5 in the proliferation of glioma cells, a series of assays, including the CCK-8, colony formation, and EdU incorporation assays, were conducted to evaluate the viability of glioma cells. The results of the CCK-8 assay demonstrated that the growth of glioma cells was significantly impeded by silencing OR7A5 compared to that in the control group, si-Con (Figs. 11B and 10C). Furthermore, colony formation assay revealed a reduction in the clonogenic ability of glioma cells following OR7A5 silencing (Fig. 11D). EdU incorporation assay also revealed a notable reduction in cellular DNA synthesis upon OR7A5 silencing (Fig. 11E). In conclusion, our findings suggested that OR7A5 enhanced the proliferative capacity of glioma cells.

## OR7A5 was associated with lipid metabolism in glioma

The correlation between OR7A5 and various signaling pathways was evaluated using ssGSEA. Notably, a positive association was observed between OR7A5 expression and glycerolipid (Fig. 12A), arachidonic acid (Fig. 12B), sphingolipid (Fig. 12C), and glycerophospholipid metabolism (Fig. 12D). These findings suggest a potential involvement of OR7A5 in lipid metabolism in gliomas. Subsequent analysis using triglyceride and total cholesterol assay kits revealed a noteworthy decrease in triglyceride and total cholesterol concentrations following the suppression of OR7A5 (Figs. 12E–12H). Western blot experiments further corroborated these findings, revealing a reduction in the expression of

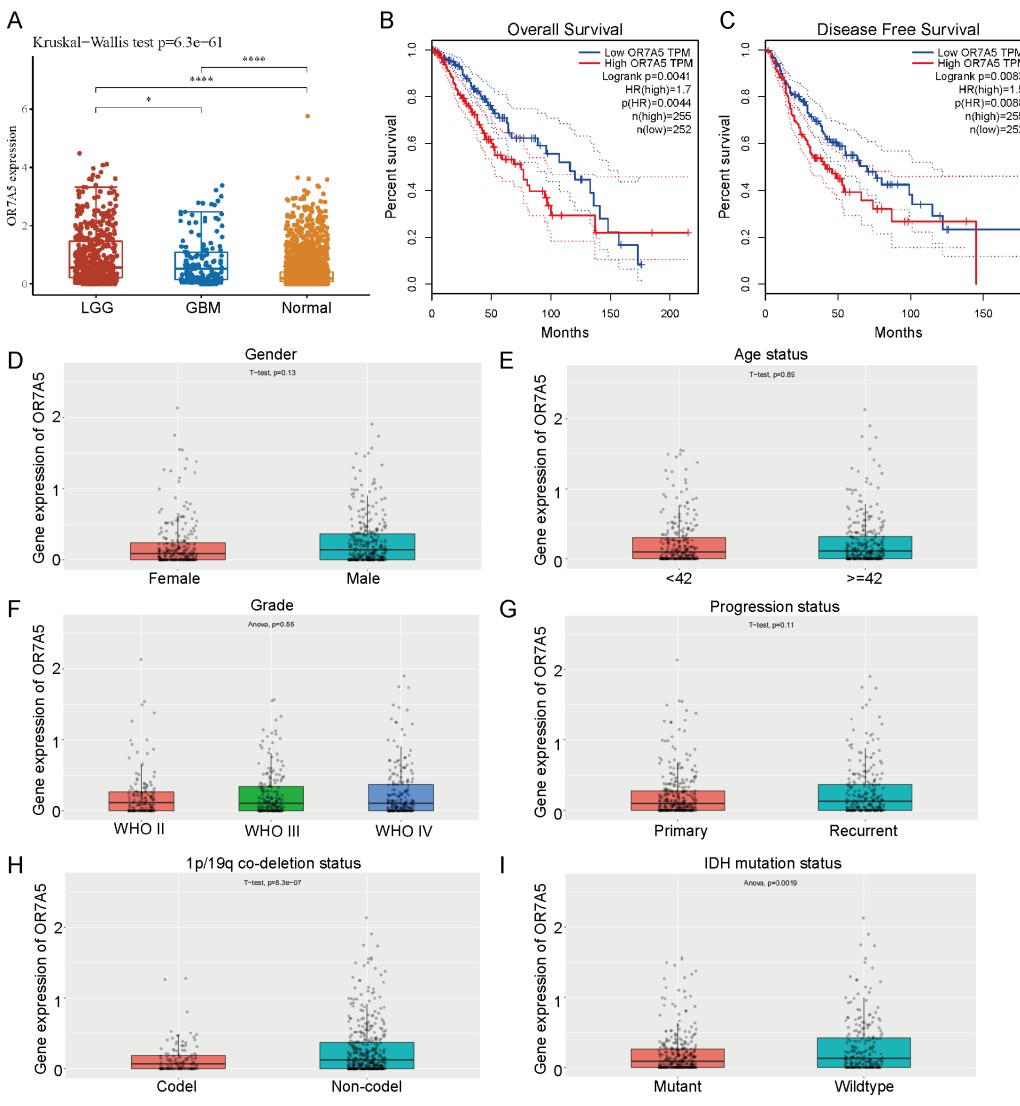

**Figure 10  Expression levels of OR7A5 in glioma and its correlation with prognosis and clinical-pathological features.** (A) Analysis of OR7A5 expression levels in LGG (G1), GBM (G2), and normal tissues through the TCGA and GTEx databases.(B) Kaplan–Meier curves of OS for subgroups with high and low OR7A5 expression. (C) Kaplan–Meier curves of DFS for subgroups with high and low OR7A5 expression. Association between OR7A5 expression levels and clinical-pathological features of patients with glioma: (D) Sex, (E) Age status, (F) Grade, (G) Progression status, (H) 1p/19q co-deletion status, (I) IDH mutation status.

pivotal lipid metabolism enzymes (FASN and SREBP1) after OR7A5 silencing (Fig. 12I). These results imply that OR7A5 facilitates glioma progression by modulating lipid metabolism.

## DISCUSSION

ORs are primarily localized in the sensory organs and play a crucial role in the detection of odors and olfaction (*Perl et al., 2020*). Increasing evidence suggests that these chemical

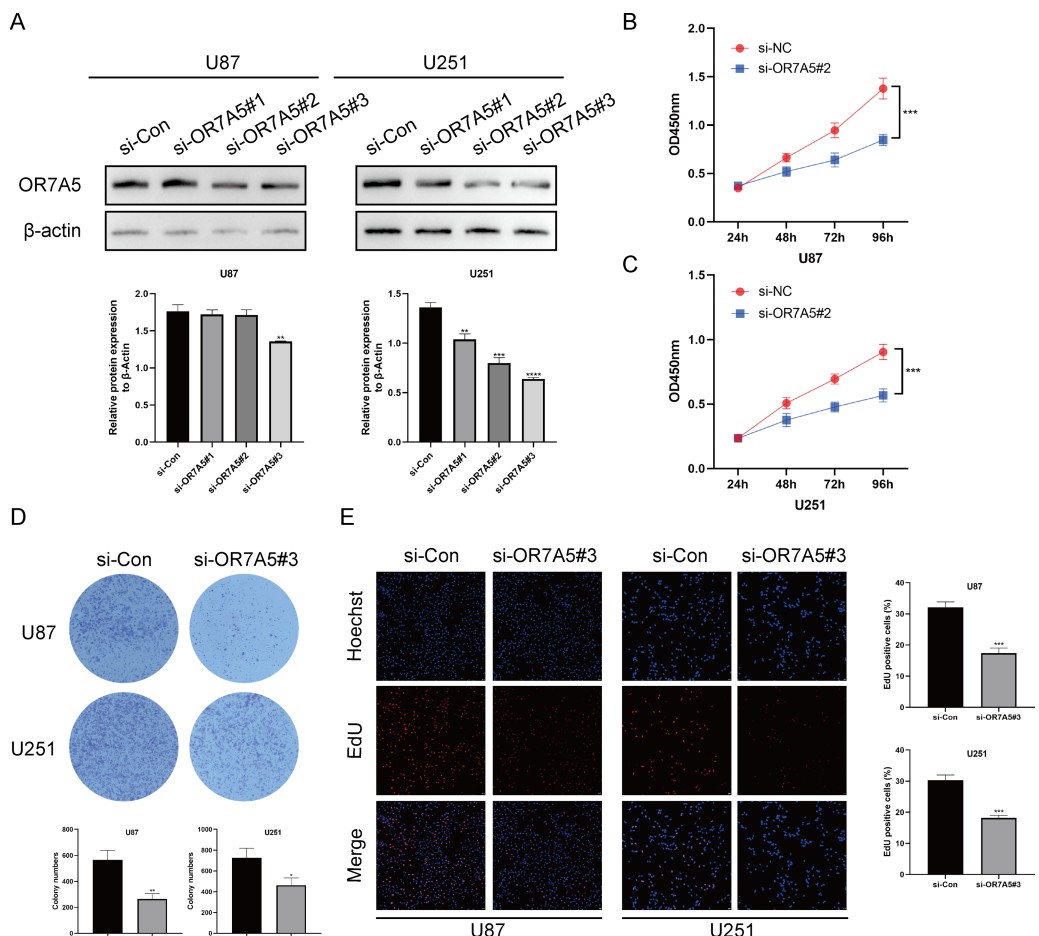

**Figure 11  Silencing OR7A5 expression inhibited proliferation ability of U87 and U251 cells.** (A) Western blot experiment to assess the silencing effect of OR7A5.(B) Impact of OR7A5 silencing on the proliferation capacity of U87 cells measured using CCK-8 assay. (C) Impact of OR7A5 silencing on the proliferation capacity of U251 cells measured using CCK-8 assay. (D) Effect of OR7A5 silencing on the clonogenic ability of U87 and U251 cells determined using colony formation assay. (E) Effect of OR7A5 silencing on the DNA replication capacity of glioma cells assessed using EdU incorporation assay. $*p < 0.05$, $**p < 0.01$, $***p < 0.001$, $****p < 0.0001$ compared to si-Con.

sensors are highly expressed in nonolfactory tissues in humans, including immune cells and tumor tissues (*Maßberg & Hatt, 2018*). OR51E2 is highly expressed in prostate cancer cells and has been implicated in the malignant progression of melanoma (*Gelis et al., 2017*; *Pronin & Slepak, 2021*). Elevated OR51E1 expression has been observed in colon, lung, and prostate cancers (*Cui et al., 2013*; *Giandomenico et al., 2013*; *Maßberg et al., 2016*). OR7C1 and OR2C3 have also been detected in colon cancer and melanoma (*Morita et al., 2016*; *Ranzani et al., 2017*). OR2AT4 has been reported to participate in the regulation of the proliferation, apoptosis, and differentiation of leukemia cells (*Manteniotis et al., 2016a*). Thes studies have underscored the potential role of ORs in cancer progression. However, the role of ORs in glioma remains unclear. Therefore, we initially utilized the TCGA database to screen for highly expressed ORs in glioma tissues, subsequently examining the

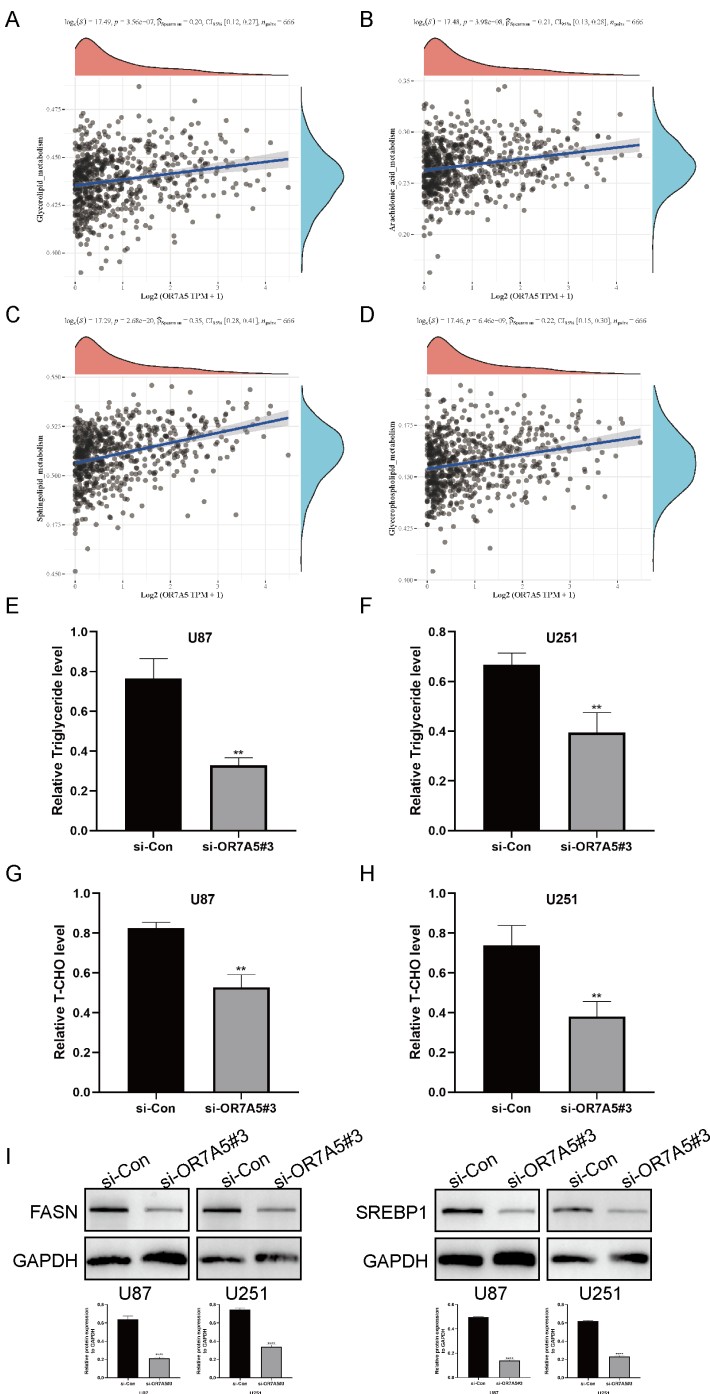

**Figure 12  Association of OR7A5 with lipid metabolism in glioma.** The correlation between OR7A5 expression levels and glycerolipid metabolism (A), arachidonic acid metabolism (B), sphingolipid metabolism (C) and glycerophospholipid metabolism (D). (E and F) Detection of changes in triglyceride levels after silencing OR7A5 using a triglyceride assay kit. (G and H) Detection of changes in total cholesterol (T-CHO) levels after silencing OR7A5 using a total cholesterol assay kit. (I) Western blot experiment to assess the expression levels of lipid metabolism-related proteins FASN and SREBP1 after silencing OR7A5. ${}^{**}p < 0.01$, ${}^{****}p < 0.0001$ compared to si-Con.

association between the expression levels of these ORs and the prognosis of patients with glioma. We observed that the expression of OR7A5 was closely linked to a poor prognosis in patients with glioma, hinting that OR7A5 may emerge as a novel oncogene, driving the malignant progression of gliomas.

Given the scarcity of research on OR7A5, we embarked on a pan-cancer analysis to delve into its impact on the genesis and development of various tumors. Our analyses, leveraging multiple public databases, revealed the upregulation of OR7A5 expression in different types of cancer. Furthermore, high OR7A5 expression correlated with a dismal prognosis in certain cancers. These findings suggest that OR7A5 plays a pivotal role in tumor initiation and progression. Analysis of DNA methylation in the OR7A5 promoter region revealed a negative correlation between OR7A5 methylation and OR7A5 expression in READ, whereas it was positively correlated with OR7A5 expression in BLCA, ESCA, UCEC, TGCT, PRAD, BRCA, LUSC, and SKCM. Survival analysis further indicated that high methylation levels of OR7A5 were associated with favorable prognosis in different cancer types. Amplification is the main cause of mRNA upregulation of OR7A5 in human cancer cells. OR7A5 is mainly involved in apoptosis, AR, PI3K/AKT, and TSC/mTOR pathways in human cancers. These findings highlighted the critical role of OR7A5 in human cancer development. As TMB and MSI have emerged as specific and sensitive biomarkers for predicting responses to immune checkpoint inhibitors, we found significant correlations between OR7A5 expression, TMB, and MSI across different cancer types. Our findings regarding the association between OR7A5 expression and immune cell infiltration, TILs, immune inhibitors, immune stimulators, MHC, chemokines, and receptors indicated that OR7A5 plays an indispensable role in modulating the immune response in human cancers.

Subsequently, we focused on elucidating the biological roles and significance of OR7A5 in gliomas. Analysis of publicly available databases highlighted a notable elevation in OR7A5 expression within glioma tissues, which notably correlated with a poorer prognosis for patients suffering from the disease. Our further exploration indicated that the expression of OR7A5 was associated with 1p/19p co-deletion status and IDH mutation. Given the specific genomic and clinical profiles that characterize various glioma subtypes, it is conceivable that distinct glioma subtypes may exhibit varied expression patterns of ORs (*Binder et al., 2019*). Prior studies have demonstrated that the enrichment of different ORs subgroups is dependent on glioma subtypes. Specifically, high expression levels of OR51E1 and OR51E2 are evident in the IDH wild-type subtype; OR4N2 is expressed in the IDH mut-non-codel subtype; and OR2L13, OR4N4, and OR4N3P are expressed at higher levels in the IDH mut-codel subtype compared to other subtypes (*Cho & Koo, 2021*). However, the heterogeneity of ORs in glioma subtypes still necessitates validation through rigorous *in vivo* and *in vitro* experiments.

To validate the oncogenic role of OR7A5 in gliomas, we conducted a series of *in vitro* experiments. These experiments revealed a significant decline in the proliferative capacity of glioma cells upon OR7A5 knockdown, indicating its crucial role in glioma growth. However, it is pertinent to note that the two cell lines we used originate from distinct backgrounds, potentially resulting in differences in protein expression and related pathways. Studies have shown disparities in protein expression between U87 and U251

cells, encompassing pathways such as nicotinamide metabolism, RNA splicing, glycolysis, and purine metabolism (*Li et al., 2017*). Therefore, future research must delve deeper into the distinct impacts of OR7A5 on U87 and U251 cells, and it is imperative to validate our findings using a broader range of cell lines to ensure a comprehensive understanding of its role in glioma.

Currently, there is a limited understanding of the signaling pathways involved in the functioning of ORs in cancer. Previous research has shown that ORs activate PKA and MAPK (*Li et al., 2019*; *Manteniotis et al., 2016b*; *Neuhaus et al., 2009*). Metabolic reprogramming is a critical regulatory mechanism for continuous tumor growth, and aberrant lipid metabolism is a significant characteristic of tumor initiation and progression (*Faubert, Solmonson & De Berardinis, 2020*; *Zhu & Thompson, 2019*). Certain ORs have been implicated in lipid regulation, specifically in the modulation of cholesterol and fatty acids (*Zhang, Li & Li, 2021*). In our current study, we observed a striking correlation between OR7A5 expression and lipid metabolism in gliomas. OR7A5 inhibition in glioma cells resulted in reduced triglyceride and total cholesterol levels, as determined using triglyceride and total cholesterol assays. Further, western blot experiments revealed a decreased expression of lipid metabolism-associated proteins, FASN and SREBP1, following the silencing of OR7A5. These findings suggest that OR7A5 plays a role in the malignant progression of gliomas by modulating lipid metabolism. Ectopic ORs are known to regulate lipid metabolism through three pathways: cAMP/HSL, cAMP/CREB, and cAMP/AMPK (*Zhang, Li & Li, 2021*). However, these mechanisms have primarily been identified in metabolic disorders and have not been implicated in tumor development. Therefore, further in-depth research is required to elucidate the involvement of ectopic ORs in lipid metabolic pathways involved in tumorigenesis.

Investigation of ORs in non-olfactory tissues is important in the academic realm. The G-protein-coupled receptor (GPCR) family, which encompasses a substantial number of approved drug targets, has garnered considerable attention, with approximately 500–700 drugs specifically designed to target GPCRs approved for clinical use (*Hauser et al., 2017*; *Sriram & Insel, 2018*). In non-olfactory tissues, ORs represent the most prominent subfamily of GPCRs, constituting approximately 60% of all human GPCRs. However, the precise functions of ORs in these tissues remain largely unknown. Consequently, exploring the involvement of the ORs family members in cancer research may yield immense potential for the development of innovative targeted clinical therapies.

However, this study has several limitations. Firstly, the research heavily depends on the analysis of public databases, and while we have cross-verified data from multiple sources, ensuring the quality and consistency of these datasets remains a challenge. Therefore, it is imperative to incorporate additional clinical glioma samples for further validation. Secondly, although we have observed the effects of OR7A5 on glioma cell proliferation *in vitro*, our investigation was confined to *in vitro* experiments. Given that *in vitro* models often fail to fully replicate the intricate physiological milieu present *in vivo*, we recognize the need for further *in vivo* experiments to authenticate our results. This will contribute to a more accurate understanding of the mechanisms underlying the role of OR7A5 in glioma development and progression. Finally, while we have noted the downregulation of

lipid metabolism-related proteins following OR7A5 inhibition, our study lacks a thorough exploration of the signaling pathways regulated by OR7A5. Specifically, the role of 12-lipoxygenase (12-LOX) and enzymes involved in ceramide and sphingosine-1-phosphate metabolism in the regulation of OR7A5 remains understudied. Hence, future research must delve deeper into the intricate interactions between these enzymes and OR7A5 to uncover the precise mechanisms of OR7A5 in glioma.

## CONCLUSIONS

In summary, this study confirmed the expression levels of OR7A5 in normal and various tumor tissues by analyzing multiple databases. Furthermore, we comprehensively analyzed the genetic and immunological characteristics of OR7A5 expression. Notably, our study presents the novel finding of upregulated OR7A5 expression in gliomas, establishing its significant correlation with adverse prognoses in patients with glioma. Additionally, we observed that the inhibition of OR7A5 resulted in decreased proliferative capacity of glioma cells, and this inhibitory effect was associated with the downregulation of the expression of lipid metabolism-related proteins. This study provides valuable insights for the further investigation of the mechanisms underlying glioma development and the potential development of innovative therapeutic approaches.

## ACKNOWLEDGEMENTS

We would like to thank Editage for English language editing.

### Funding

This work was supported by the Shaoxing Municipal Science and Technology Plan Project of China under Grant (No. 2022A14022) and the Shaoxing Health Science and Technology Project of China under Grant (No. 2022KY003). The funders had no role in study design, data collection and analysis, decision to publish, or preparation of the manuscript.

### Grant Disclosures

The following grant information was disclosed by the authors:
Shaoxing Municipal Science and Technology Plan Project of China: 2022A14022.
The Shaoxing Health Science and Technology Project of China: 2022KY003.

### Competing Interests

The authors declare there are no competing interests.

### Author Contributions

- Yanqiu Bao performed the experiments, analyzed the data, prepared figures and/or tables, authored or reviewed drafts of the article, and approved the final draft.
- Ziqi Tang performed the experiments, analyzed the data, prepared figures and/or tables, authored or reviewed drafts of the article, and approved the final draft.

- Renli Chen performed the experiments, prepared figures and/or tables, and approved the final draft.
- Xuebin Yu conceived and designed the experiments, authored or reviewed drafts of the article, and approved the final draft.
- Xuchen Qi conceived and designed the experiments, authored or reviewed drafts of the article, and approved the final draft.

## Data Availability

Raw data are available in the Supplemental Files.

## Supplemental Information

Supplemental information for this article can be found online at http://dx.doi.org/10.7717/peerj.17631#supplemental-information.

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
