# Peer review of "Pan-cancer analysis identifies olfactory receptor family 7 subfamily A member 5 as a potential biomarker for glioma"

_PeerJ, doi:10.7717/peerj.17631_

## Round 0.1 · original submission · Major Revisions

Thank you for submitting your manuscript, "Pan-cancer analysis identifies olfactory receptor family 7 subfamily A member 5 as a potential biomarker for glioma," to PeerJ for consideration. We appreciate the time and effort you have invested in this research.

After careful review by our expert panel, we have reached a consensus decision on your manuscript. We are pleased to inform you that your work has been deemed potentially suitable for publication in PeerJ, but it requires significant revision before it can be accepted.

The reviewers have raised several key points that need to be addressed in a major revision. Primarily, they have emphasized the need for a more robust explanation of why OR7A5 was chosen as the focus of your study. A clearer rationale for this selection would greatly enhance the impact and credibility of your findings.

Additionally, the reviewers have commented on the clarity of the figures and charts. It is essential that these visual elements are as clear and informative as possible, as they play a crucial role in communicating your research findings. Please ensure that all figures and charts are revised to address the reviewers' concerns and enhance their overall readability.

We also encourage you to carefully consider and respond to all other comments and suggestions made by the reviewers. Their feedback is intended to help you strengthen your manuscript and make it more suitable for publication.

When submitting your revision, please ensure that you have addressed all the concerns raised by the reviewers and included a point-by-point response to their comments. This will aid us in the evaluation of your revised manuscript.

We look forward to receiving your revised manuscript and continue working with you towards its publication in PeerJ. If you have any questions or need assistance during the revision process, please feel free to contact us at any time.

·

Basic reporting

Pan-cancer analysis identifies olfactory receptor family 7 subfamily A member 5 as a potential biomarker for glioma

by Yanqiu Bao et al.

Performed a comprehensive bioinformatics pancancer analysis of te expression of the OR7A5 olfactory receptor supplemented by experimental views in cell lines. The authors report that OR7A5 expression associates with adverse prognosis, 1p19q codeletion and IDH-wt in gliomas and therefore represents a potential diagnostic marker and target for treatment.

G-protein coupled receptors and particularly olfactory receptors are understudied in cancers although their function can associate with tumor development and behavior.
In this sense the study of OR7A5 using a battery of bioinformatics methods and data fills a gap and provides interesting insights into the transcriptomics across many cancer types, associate them with biological functions and clinical data with the focus on gliomas.
Overall the study is technically well done and provides a series of interesting details.

I support the intention of the manuscript and the presentation of the results as publication in PeerJ.

My evaluation is contradictionary: On one hand the material is overwhelming and rich in facts and as such valuable for specialists. On the other hand, the results are not really digested and presented in a very sparse style, what leaves the reader in a “so what” situation. The story line is weak. It needs discussion of the different effects and their mutual evaluation and interpretation. The quality of figures is also of this yin-yan style: Overall okay, but partly with unreadable marks and partly difficult to understand contents.

The story needs considerable improvement. Might a “less is more” could be a good advice? Particularly is needs a rationale regarding cancer diversity and types- genomic and clinical features- biological background. If the authors could link those points at least partly, the work would considerably gain in impact.

My points are:

1.) It is not clear, why the authors selected particularly OR7A5 and not any other olfactory receptor? The claim “for the first time”…well its not a great deal to pick one particular OR gene from a long list. It would be more constructive to put the possible impact of PanCancer studies for ORs in general and then to argue what OR7A5 will contribute to this issue.
2.) Lines 210-240 and Fig.3: The labels in fig.3 are partly too small and (nearly) unreadable. The descriptions are are partly a bit sparse, e.g. the context of differential expression between subtypes (Fig 3F). Also the pathway enrichment needs a bit more words, particularly is the rationale of this. What does Fig. 3E show? What pathways are enriched. This Figure is not understandable to me? Finally, it is mentioned that “protein expression” is affected, but the authors show “gene expression”?
3.) Line 243ff and Fig. 4: Again, labels in the figure are too small (applies to nearly all figures). What do the left and the right figures on LGG show? What is the difference between the two data bases/methods. The “ACC” label should be similar to the resp. labels of the other cancers. Legend: Part B shows Kaplan-Meyer curves and no correlations. Overall figures have to be improved in these respects. Descriptions in the results section are very sparse and leave the reader a bit helpless: One sees that nature is complicated, so what?
4.) Genetics, line 256ff and the other subsections in the results part: the same critique as for expression: a lot of data but (nearly) no rationale and very sparse comments. Figure is unreadable. Wg´hy, e.g. in Fig. 6 the authors show OS, PSF and DSS…what are the differences and what is the rationale to show them.

5.) General about results: It looks like a detailed and elaborated report about comprehensive bioinformatics analysis results without concept, explanation and interpretation. Can the authors summarize the major rationale of each of the subsections to guide the reader along a sort of red line. I don’t see a story line.

6.) The authors focus then partly on Gliomas (GBM and LGG) what is an interesting topic because DNA methylation of OR genes (eg on Chr. 11) is strongly affected in glioma subtypes but almost not understood (see eg PMID: 31023364). Moreover, OR effects across glioma subtypes is very heterogeneous and partly associates with IDH mutation and 1p19q Chr. status as the authors mentioned. The association with lipid metabolism is an interesting result. But again: Results are not really understood and need discussion on the backround with literature data …

Experimental design

see basic review for details. Overall okay. Critique: too much and partly not understandable. Also the reason why one particular gene is selected is not clear.

Validity of the findings

Interesting and valuable, see basic report (no 1). However, its more a description of an overwhelming number of different effects, which needs improvement in terms of a general discussion.

Additional comments

Olfactory receptos are understudied and nor really understood in the cancer context. I very appreciate the efforts of the authors to address this problems. I dont expect comprehensive answers on all issues. The paper provides a rich material whic, however, more decscribes the problem. I strongly recommend to extend and to improve the story and better to describe the results.

Reviewer 2 ·

Basic reporting

Overall, the paper is well-written and straightforward. The flow chart of the study while unnecessary, was a nice figure, though it would benefit the readers if the authors could elaborate more on the salient features of the databases used and that made the study possible. Perhaps include a comparative table of databases with pros and cons of each listed (e.g., immune infiltrates are inferred from the literature, but representation of other aspects of the microenvironment that may contribute to OR7A5 are lacking, etc.). There are many acronyms throughout and while these are adequately described as required by the style requirements, it made reading the article cumbersome. Magnification of figure axes will be problematic in a non-digital format as well as the contrast of Figure 10E. That could be improved.

Experimental design

The authors used public databases to examine 5 parameters related to cancer with respect to OR7A5, namely (i) patterns of expression, (ii) prognostic correlations, (iii) DNA methylation, (iv) genetic and epigenetic features, and (v) immune infiltration (e.g., TISIDB and TIMER2 databases). The authors also performed a glioma-specific, targeted comparison in silico of OR7A5 expression with clinicopathological characteristics of glioma patients utilizing the Chinese Glioma Genome Atlas (CGGA). There appears to be an association between overall survival and OR7A5 in several cancer types and all 5 parameters of the OR that were assessed (mentioned above) demonstrated both favorable and adverse outcomes depending on the cancer type.

In silico findings and potential modes of action were validated experimentally in vitro using U87 and U251 authenticated human glioma cell lines from a national repository in Shanghai, China. Features of these cell lines should be considered in the discussion of results as U87 is considered neuronal and more proliferative than U251, which is thought to be more mesenchymal and slower growing. In vitro methods included OR7A knockdown by siRNA transfection and subsequent assays for cell viability, colony formation and EdU incorporation. Additional analyses were performed to examine lipid metabolism and the effect of OR7A5 suppression on levels of proteins related to lipid metabolism, namely FASN and SREBP1.

Regarding experimental studies, the authors examined several prominent bioactive lipid classes, namely arachidonic acid, sphingolipids, etc. However, as an example, it is well known that the lipoxygenase-generated arachidonic acid metabolite 12(S)-HETE factors prominently in many cancers. Therefore, it would have been important and highly relevant to evaluate the platelet type 12-lipoxygenase and other enzymes of this class for their responsiveness/connection to the signaling pathways being regulated by OR7A, particularly as systemic administration of specialized pro-resolving mediators has been shown to mitigate some of the adverse effects of these proinflammatory lipids and may prove useful in glioma treatment. Similarly, receptors/enzymes associated with ceramide and sphingosine-1-phosphate metabolism could have been evaluated. Perhaps the authors plan this for future experiments, but this should at least be mentioned in the Discussion in more detail since the experiments were not performed and since these may in/directly link OR75A to glioma.

Validity of the findings

The authors used public databases to examine 5 parameters related to cancer with respect to OR7A5, namely (i) patterns of expression, (ii) prognostic correlations, (iii) DNA methylation, (iv) genetic and epigenetic features, and (v) immune infiltration (e.g., TISIDB and TIMER2 databases). The authors also performed a glioma-specific, targeted comparison in silico of OR7A5 expression with clinicopathological characteristics of glioma patients utilizing the Chinese Glioma Genome Atlas (CGGA). There appears to be an association between overall survival and OR7A5 in several cancer types and all 5 parameters of the OR that were assessed (mentioned above) demonstrated both favorable and adverse outcomes depending on the cancer type.

In silico findings and potential modes of action were validated experimentally in vitro using U87 and U251 authenticated human glioma cell lines from a national repository in Shanghai, China. Features of these cell lines should be considered in the discussion of results as U87 is considered neuronal and more proliferative than U251, which is thought to be more mesenchymal and slower growing. In vitro methods included OR7A knockdown by siRNA transfection and subsequent assays for cell viability, colony formation and EdU incorporation. Additional analyses were performed to examine lipid metabolism and the effect of OR7A5 suppression on levels of proteins related to lipid metabolism, namely FASN and SREBP1.

Additional comments

The investigators have chosen to explore in more detail the role of olfactory receptors (ORs) in non-native functions, specifically in development and advancement of various cancers with a focus on glioblastoma where novel biomarkers could improve diagnosis and treatment efficacy.
The idea for a comparative study to characterize ORs between cancers is rooted in observations by others that ORs (e.g., OR51E2 in prostate cancer and melanoma) are upregulated in areas other than the nasal cavity such as smooth muscle, skin, and hair follicles. Additional studies have also suggested that olfactory dysfunction may serve as a prognostic factor in glioblastoma. This reviewer was unable to locate the rationale for choosing to focus specifically on OR7A5.

Reviewer 3 ·

Basic reporting

Pros:
The manuscript is well-structured, following conventional scientific report format.
A transparent background is provided that contextualizes the research within the broader field of tumor biology and olfactory receptors.
Figures and tables are appropriately used, enhancing the presentation of results.
The authors have provided comprehensive references, demonstrating a thorough understanding of the relevant literature.

Cons:
While the language is generally clear, some sections could benefit from minor grammatical and syntactical revisions for improved readability.

Experimental design

1. Relying on existing databases raises questions about potential variability in data quality and consistency. A discussion of how these factors were mitigated would be valuable.
2. The study relies heavily on secondary data, and primary experimental validation is limited to in vitro assays. In vivo studies or clinical samples would significantly strengthen the conclusions.
3. The authors should discuss their study's limitations more thoroughly, especially the potential biases inherent in using multi-source databases.

Validity of the findings

N/A

Additional comments

The study presents a valuable contribution to the field of cancer biomarker research. However, its reliance on database analysis, while methodologically sound, could be complemented with more experimental data, especially from clinical samples or in vivo studies.
The potential translational impact of these findings is significant, and the authors might consider discussing the pathway toward clinical application in more depth, including the challenges and potential strategies.
The manuscript could benefit from a more detailed discussion on how OR7A5 could be practically utilized in clinical settings for glioma diagnosis or therapy.
The paper's focus on a pan-cancer analysis is commendable, but future studies might need to delve deeper into specific cancer types to fully understand the role of OR7A5 across different tumor environments.

---

## Round 0.2 · accepted · Accept

This manuscript can be accepted now.

·

Basic reporting

The authors addressed all my points appropriately. For me the paper is fine now. Congrats to the authors.

Experimental design

No further comments.

Validity of the findings

No further comments.

Additional comments

None.

Reviewer 3 ·

Basic reporting

No comment

Experimental design

No comment

Validity of the findings

No comment

Additional comments

No comment